# Incoherent nonadiabatic to coherent adiabatic transition of electron transfer in colloidal quantum dot molecules

Bokang Hou [1], Michael Thoss[2], Uri Banin [3] & Eran Rabani [1,4,5] ✉

Electron transfer is a fundamental process in chemistry, biology, and physics. One of the most intriguing questions concerns the realization of the transitions between nonadiabatic and adiabatic regimes of electron transfer. Using colloidal quantum dot molecules, we computationally demonstrate how the hybridization energy (electronic coupling) can be tuned by changing the neck dimensions and/or the quantum dot sizes. This provides a handle to tune the electron transfer from the incoherent nonadiabatic regime to the coherent adiabatic regime in a single system. We develop an atomistic model to account for several states and couplings to the lattice vibrations and utilize the mean-field mixed quantum-classical method to describe the charge transfer dynamics. Here, we show that charge transfer rates increase by several orders of magnitude as the system is driven to the coherent, adiabatic limit, even at elevated temperatures, and delineate the inter-dot and torsional acoustic modes that couple most strongly to the charge transfer dynamics.

The theoretical framework to describe charge transfer reactions in condensed phases dates back to the seminal work of Marcus[1], where he considered a donor molecule weakly coupled to an acceptor molecule, and developed a theoretical framework to describe the electron transfer reactions in a fluctuating environment. Using a semi-classical perturbative approach, Marcus derived a relation for the outer-shell electron transfer rate at high temperatures ($T$) in terms of the driving force ($\Delta\varepsilon$), the reorganization energy ($\lambda$) characterizing the coupling to nuclear fluctuations, and the hybridization (electronic couplings) between donor and acceptor states ($J$), assumed to be small (see Fig. 1a for a sketch of the relevant energy scales):

$$k_M = \frac{|J|^2}{\hbar}\sqrt{\frac{\pi}{\lambda k_B T}}\exp\left\{\frac{-(\Delta\varepsilon + \lambda)^2}{4\lambda k_B T}\right\}. \tag{1}$$

Marcus theory is suitable for the nonadiabatic electron transfer regime, where the electron transfer from the donor to the acceptor can be viewed as a nonadiabatic transition between the diabatic donor and acceptor states (see dashed curves in Fig. 1a). This nonadiabatic limit is characterized by a small value of the so-called "adiabatic parameter", $\gamma$, defined as the ratio between the donor-acceptor hybridization ($J$), the characteristic nuclear vibrational frequency ($\omega_c$), and the reorganization energy $\lambda$[2]:

$$\gamma = \frac{|J|^2}{2\hbar\omega_c}\sqrt{\frac{\pi}{\lambda k_B T}}, \tag{2}$$

The nonadiabatic electron transfer Marcus regime is thus achieved either for weak electronic coupling and/or for fast nuclear motion ($\omega_c > J/\hbar$).

Marcus nonadiabatic electron transfer theory was extended in several different directions. Jortner and coworkers described the role of quantum nuclear fluctuations as well as non-parabolicities in the donor and acceptor free energies on the electron transfer rate[3]. Redfield theory[4] extended Marcus theory to account for coherences between the donor and acceptor, observed in photosynthetic

[1]Department of Chemistry, University of California, Berkeley, CA 94720, USA. [2]Institute of Physics, University of Freiburg, Hermann-Herder-Straße 3, 79104 Freiburg, Germany. [3]Institute of Chemistry and the Center for Nanoscience and Nanotechnology, The Hebrew University of Jerusalem, 91904 Jerusalem, Israel. [4]Materials Sciences Division, Lawrence Berkeley National Laboratory, Berkeley, CA 94720, USA. [5]The Raymond and Beverly Sackler Center of Computational Molecular and Materials Science, Tel Aviv University, 69978 Tel Aviv, Israel. ✉e-mail: eran.rabani@berkeley.edu

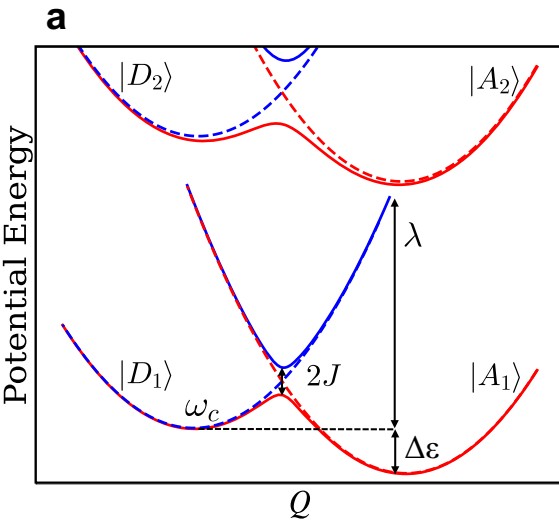

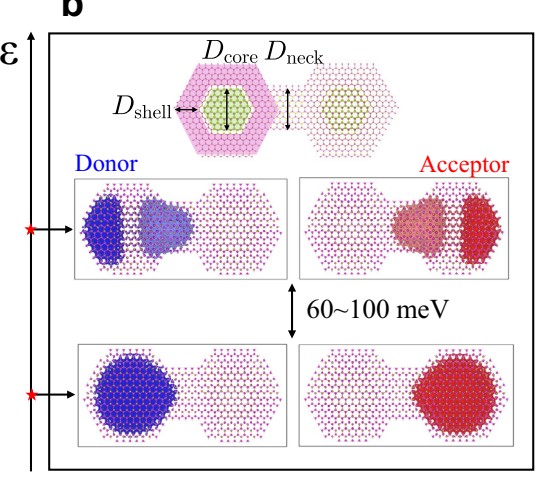

**Fig. 1 | Illustration of donor and acceptor states in the electron transfer model.** **a** Schematic sketch of the potential energy surfaces along the reaction coordinate $Q$ for the nonadiabatic electron transfer and the energy scales appearing in Eq. (1): driving force $\Delta\varepsilon$, donor-acceptor hybridization $J$, reorganization energy $\lambda$, and characteristic frequency $\omega_c$. The dashed lines colored blue (donors, $|D_1\rangle, |D_2\rangle \ldots$) and red (acceptors, $|A_1\rangle, |A_2\rangle \ldots$) represent diabatic potential energy surfaces. The solid lines represent adiabatic potential energy surfaces. **b** The dimensions of the core-shell wurtzite colloidal quantum dot (CQD) dimers are characterized by the neck width $D_{neck}$, CdSe core diameter $D_{core}$, and CdS shell thickness $D_{shell}$. Hyper-sphere plots of the donor (left nanocrystal, blue) and acceptor states (right nanocrystal, red) were obtained from the semi-empirical pseudopotential calculation and FB-localization of the quasi-electron eigenstates. The dark and light colors of the wavefunctions indicate positive and negative phases, respectively. The energy difference between $1S_e$-like ($|D_1\rangle$, $|A_1\rangle$) and $1P_e$-like ($|D_2\rangle$, $|A_2\rangle$) orbitals is in the range of 60–100 meV, depending on the size of the nanocrystal building blocks.

complexes and quantum dot islands exhibiting strong electronic couplings[5,6]. Unlike the Marcus regime in which the transfer dynamics decay exponentially and can be characterized by a rate constant, the population dynamics in this coherent limit oscillate between the donor and acceptor, eventually relaxing to equilibrium.

Zusman developed a framework to describe the crossover from the Marcus weak coupling nonadiabatic limit to the adiabatic limit[7,8], where the coupling between donor and acceptor is large and/or the nuclei motion is slow ($\gamma \gg 1$). In the adiabatic limit, diabatic states are no longer a good representation and the dynamics proceed on a single adiabatic (Born-Oppenheimer) surface (see solid lines in Fig. 1a). The pre-exponential factor appearing in Marcus' rate theory (c.f., Eq. (1)) becomes implicitly dependent on the hybridization between the donor and the acceptor, and the rate constant is given by transition state rate theory:

$$k_{TST} = \frac{\omega_c}{2\pi}\sqrt{\frac{\pi\lambda}{k_B T}}\exp\left\{\frac{-(\Delta\varepsilon + \lambda)^2}{4\lambda k_B T}\right\}. \quad (3)$$

Experimental manifestation of transition from nonadiabatic to adiabatic charge transfer is challenging and requires exquisite control over electronic and vibrational degrees of freedom[9,10]. Recently, Zhu et al.[11] studied electron transfer reactions in mixed valance donor-bridge-acceptor complexes. By changing the length of the bridge connecting the donor and acceptor and the functional groups on both the donor and the acceptor, they were able to drive the system from the overdamped adiabatic to the nonadiabatic regime. However, their molecular systems were limited to either 1, 2, or 3 bridge units and in the strong coupling limit to the solvent, covering a narrow range of electron transfer behaviors.

In this work, we revisit this problem and consider the charge transfer between two coupled colloidal quantum dot (CQD) nanocrystals (NCs) that are connected by a neck/bridge (see Fig. 1b). Charge transfer in such systems is particularly interesting due to the flexibility in designing the donor and acceptor states by, for example, changing the width of the neck ($D_{neck}$) between two NCs and/or the diameter of

each NC core ($D_{core}$), as well as controlling the shell thickness ($D_{shell}$)[12–14]. By continuously varying these parameters, the hybridization between the donor and the acceptor, $J$, can be tuned across a wide range of values while at the same time the reorganization energy, $\lambda$, and the typical vibrational frequency, $\omega_c$, change slightly.

In addition, the hybridization energies can be tuned to be larger than the thermal energy at room temperature without affecting the reorganization energy. This is quite distinct from the behavior of molecular junctions, where a change in the hybridization energy is often accompanied by a change in the reorganization energy and the vibrational frequency. This is because control over these parameters in molecular junctions is achieved by either changing the donor/acceptor molecules or by extending the bridge connecting them, which results in changes in the other parameters as well. The flexibility of controlling the hybridization energy without affecting the other energy scales in the system, offers a platform to drive the system from the overdamped electron transfer dynamics typical to molecular systems to the coherent limit, where the electron transfer rates are governed by decoherence times, even at elevated temperatures.

## Results and discussion
### Model Hamiltonian
From a theoretical/computational perspective, studying electron transfer in a coupled CQD system poses several challenges, particularly with respect to the dimensions and number of valance electrons. Therefore, our approach to describing electron transfer is based on a model Hamiltonian, which is parameterized by first-principle and semi-empirical calculations. The approach was recently validated in predicting optical gap, reorganization energies/Stokes shifts, linear optical PL spectrum, multi-excitonic effects, and more[14–18]. The total Hamiltonian can be divided into a sum of three terms, $H = H_S + H_B + H_{SB}$, where $H_S$ describes the electronic system (donor and acceptor states and their hybridization), $H_B$ is the Hamiltonian for the nuclear degrees of freedom (DOF) of the QD dimer (nuclear vibrations), and the interaction between the electronic system and the nuclear vibrations, approximated to the lowest order in the nuclear

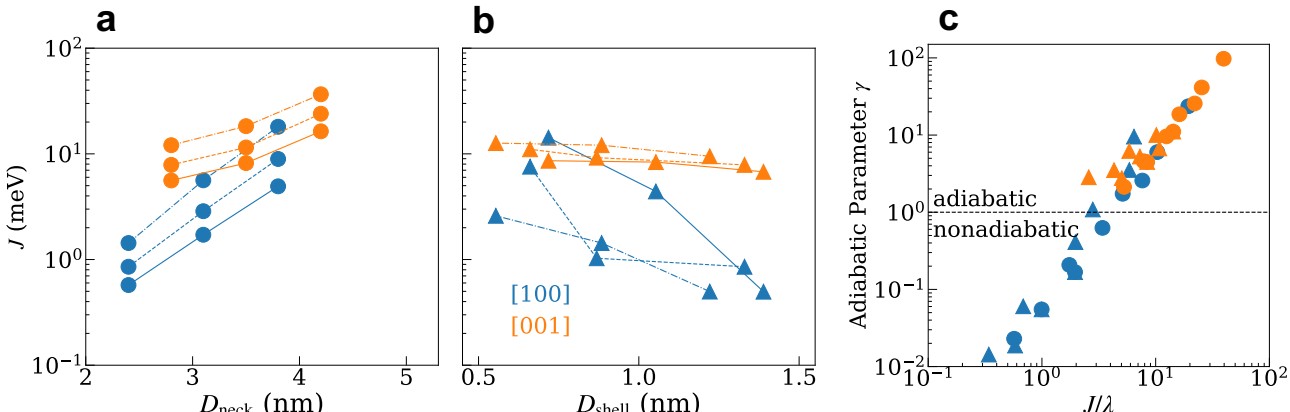

**Fig. 2 | Electronic properties of the QD dimers. a, b** Hybridization energy $J$ of the ground donor and acceptor states as a function of neck width $D_{neck}$ (**a**) and shell thickness $D_{shell}$ (**b**) for [100] and [001] orientation attachments. Circle and triangle represent controlling the QD dimer by $D_{neck}$ and $D_{shell}$, respectively. Solid, dashed, and dotted-dashed lines correspond to core diameters $D_{core}$ of 2.2 nm, 3.0 nm, and 3.9 nm. **c** Adiabatic parameter $\gamma$ vs. ground state hybridization/reorganization energy $J/\lambda$ for varying neck-core sizes or shell-core sizes at 300 K. We define dashed line $\gamma = 1$ as the crossover from the nonadiabatic to adiabatic electron transfer regimes. Source data are provided as a Source Data file.

DOF[19], is described by $H_{SB}$. The three terms are given by:

$$H_S = \sum_{n \in \mathcal{D},\mathcal{A}} \varepsilon_n |\phi_n\rangle\langle\phi_n| + \sum_{\substack{n \in \mathcal{D} \\ m \in \mathcal{A}}} J_{nm} |\phi_n\rangle\langle\phi_m| + h.c. \quad (4)$$

$$H_B = \sum_\alpha \frac{1}{2} P_\alpha^2 + U(Q_1, Q_2, \ldots) \quad (5)$$

$$H_{SB} = \sum_\alpha \sum_{n,m \in \mathcal{D},\mathcal{A}} |\phi_n\rangle\langle\phi_m| V_{nm}^\alpha Q_\alpha. \quad (6)$$

In the above equations, $\varepsilon_n$ is the energy of state $|\phi_n\rangle$ ($n \in \mathcal{D},\mathcal{A}$ with $\mathcal{D} = \{D_1, D_2 \ldots\}$, $\mathcal{A} = \{A_1, A_2 \ldots\}$) and $J_{nm}$ is the hybridization between the donor state $|\phi_n\rangle$ and acceptor state $|\phi_m\rangle$. To obtain the donor and acceptor states, we used the semi-empirical pseudopotential model[20,21] to describe the quasi-electron Hamiltonian ($\hat{h}_{QP}$) and the filter-diagonalization technique[22,23] to calculate the eigenstates ($|\psi_i\rangle$) of the dimer near the bottom of the conduction band. The Förster-Boys localization scheme[24,25] was then used to generate the local donor and acceptor states ($|\phi_n\rangle$) from the eigenstates ($|\psi_i\rangle$), yielding $\varepsilon_n = \langle\phi_n|\hat{h}_{QP}|\phi_n\rangle$ and $J_{nm} = \langle\phi_n|\hat{h}_{QP}|\phi_m\rangle$ for $n \in \mathcal{D}, m \in \mathcal{A}$, and otherwise by construction, it is set to $J_{nm} = 0$. We use the Stilinger-Weber[26] potential energy surface to describe the nuclear degrees of freedom, where $P_\alpha$ and $Q_\alpha$ are the $\alpha$ mass-weighted vibrational normal mode coordinates, determined by diagonalizing the Hessian matrix at the equilibrium geometry[19]. The strength of coupling between states $|\phi_n\rangle$ and $|\phi_m\rangle$ to mode $\alpha$ is given by $V_{nm}^\alpha$, and is determined directly using the pseudopotential Hamiltonian[19] (see Eq. (14) in Methods).

Several low-lying donor and acceptor states for the attachment orientation [100] are shown in Fig. 1b. The [100] attachment results in a symmetric distribution of the charge density, similar to a homonuclear diatomic molecule, with an "atomic-like" basis of an effective mass particle-in-a-sphere model. The two lowest-energy donor/acceptor states are mainly comprised of $1S_e$-like orbitals, while higher-lying states show $1P_e$-like character (either along or perpendicular to the dimer axis). For the systems considered in this work, the energy gap between $1S_e$-like and $1P_e$-like orbitals ranges from 60 to 100 meV. The symmetry is broken along the [001] orientation attachment, resulting in a small energy bias and an asymmetric charge distribution, consistent with the behavior of hetero-nuclear diatomic molecules (see Supplementary Figs. 1 and 2 for details).

In Fig. 2a, b we plot the hybridization energy between the ground donor and acceptor states, $J = J_{D_1 A_1}$, as a function of the neck width ($D_{neck}$) and shell thickness ($D_{shell}$), respectively, with different NC core diameters ($D_{core}$). As expected, increasing the neck widths and core diameters or decreasing the shell thickness, results in an exponential increase of the magnitude of the hybridization energy. Furthermore, the [100] attachment (blue curves in Fig. 2a, b) shows a much steeper dependence on $D_{neck}$ and $D_{shell}$, as a result of the larger core-to-core distance for this orientation. In Fig. 2c we plot the adiabatic parameter, $\gamma$, as a function of the hybridization energy for the same set of neck and shell dimensions shown in Fig. 2a, b. We find a crossover from the nonadiabatic to the adiabatic electron transfer regimes as $J \to \lambda$. Since the characteristic vibrational frequency and the reorganization energy depend weakly on the dimer geometry, we find that the crossover is mainly affected by the hybridization energy. Note that for this set of building block monomers, the electron transfer in the [001] orientation attachment is in the adiabatic limit regardless of the neck width or shell thickness, while it can be tuned from the nonadiabatic to the adiabatic limits for the [100] orientation attachment. The detailed dimensions of all QD dimers shown in Fig. 2 are summarized in the Supplementary Tables 1 and 2.

## Electron transfer dynamics in quantum dot dimers

In Fig. 3 we plot the donor and acceptor populations as a function of time in the nonadiabatic-Marcus regime ($\gamma \ll 1$), the intermediate regime ($\gamma \approx 1$), and the adiabatic regime ($\gamma \gg 1$). We used the Ehrenfest mean-field mixed quantum-classical method[27–29] to describe the dynamics in all three regimes. As discussed below (c.f., Fig. 4), the electron is mainly coupled to the low-frequency acoustic modes, for which the classical limit is adequate ($\hbar\omega_c \ll k_B T$). For $\gamma \ll 1$ in Fig. 3a we also compare the mean-field results to a master equation (due to the presence of multiple donor and acceptor states) with transition rates obtained from Marcus theory. The donor population $p_\mathcal{D}(t)$ and acceptor population $p_\mathcal{A}(t)$ are given as the projections onto the donor and acceptor Hilbert spaces, respectively (see Eq. (19) in Methods). Individual state populations corresponding to the results shown in Fig. 3 are shown in Supplementary Figure 3 (population dynamics for all other structures studied in this work are shown in Supplementary Figure 4 to 7).

For weak hybridization between donor and acceptor states (i.e., for small $D_{neck}$ and/or large $D_{shell}$), the population dynamics are characterized by an over-damped exponential decay shown in Fig. 3a, with a decay rate that approximately matches the Marcus rate between the

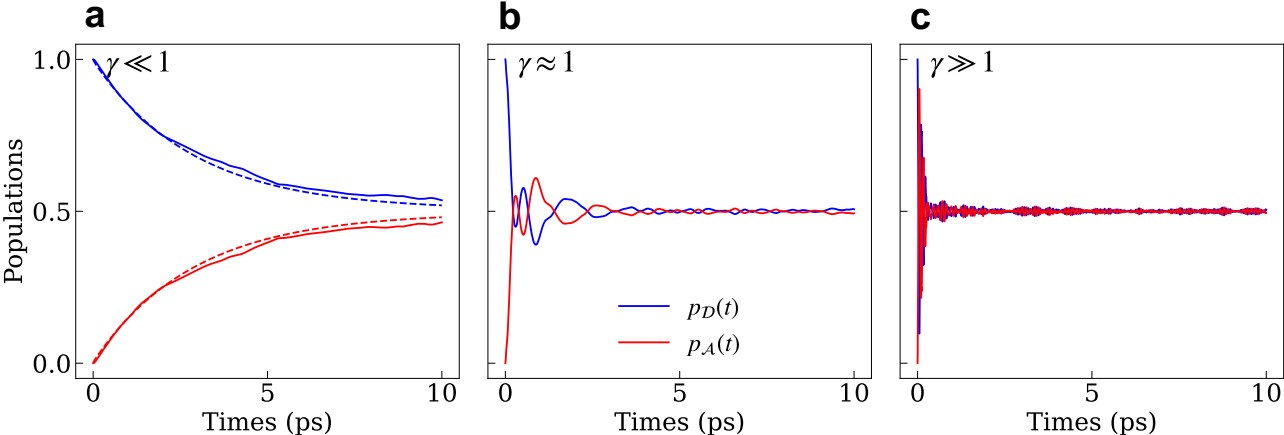

**Fig. 3 | Population dynamics of electron transfer at 300 K for different regimes.** **a**–**c** Donor population $p_D(t)$ and acceptor population $p_A(t)$ in the nonadiabatic (**a**), intermediate (**b**), and adiabatic (**c**) regimes, characterizing by the adiabatic parameter $\gamma$. All structures are in the symmetric attachment, leading to $p_D(\infty) = p_A(\infty) = 0.5$ in the long-time limit. The dashed lines in **a** are the population generated from the master equation with rates computed using Marcus theory. Source data are provided as a Source Data file.

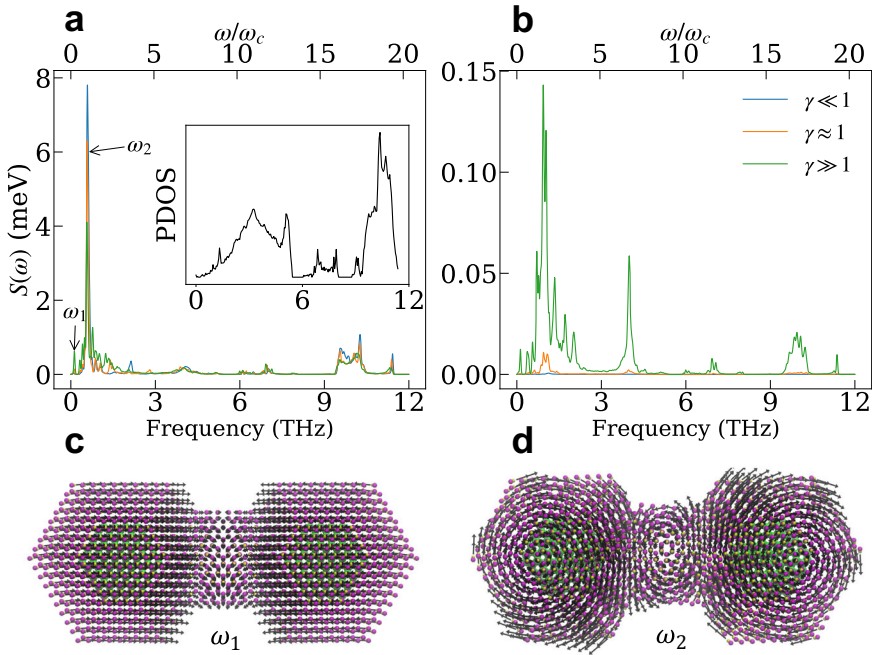

**Fig. 4 | Vibronic properties of QD dimers. a, b** Diagonal spectral densities $S_{D_1D_1}(\omega)$ (**a**) and off-diagonal spectral densities $S_{D_1A_1}(\omega)$ (**b**) for the same systems characterized by the adiabatic parameter $\gamma$ as in Fig. 3. The blue, orange and green lines correspond to spectral densities of the nonadiabatic, intermediate, and adiabatic regimes as shown in Fig. 3 panel **a**–**c**. The inset in **a** shows the phonon density of state (PDOS). **c, d** The two most important modes contributing to dephasing are the acoustic inter-dot vibrational mode with $\omega_1 = 0.13$ THz (**c**) and the torsional mode with $\omega_2 = 0.58$ THz (**d**). The arrows indicate the motion of each atom for each mode. Source data are provided as a Source Data file.

ground donor and acceptor states. For intermediate values of the shell and neck thicknesses, i.e., for $\gamma \approx 1$, the population dynamics show underdamped coherent oscillations, with a Rabi frequency that matches predominately the ground state donor-acceptor hybridization (Fig. 3b). As the neck width is further increased and/or the shell thickness is further decreased, the Rabi period shortens, signifying the increase in the hybridization energy, as shown in Fig. 3c.

The agreement between the mean-field theory and the perturbative Marcus result for $\gamma \ll 1$ is consistent with the so-called "average classical limit"[30], where the dynamics of the nuclear degrees of freedom are governed by the arithmetic average donor/acceptor

Hamiltonian[31]. The average classical limit has been motivated by the analysis of the Wigner form of the quantum mechanical expression for the relevant time-correlation function[32], suggesting that the average Hamiltonian (similar to mean field) provides the most accurate approximation to the fully quantum mechanical results[31]. For large values of $\gamma$, where the hybridization energy is much larger than the reorganization energy ($J \gg \lambda$) and the system is in the weak electron-phonon coupling limit, the average mean force on the nuclei is similar to the force for each diabatic potential energy surface, and the mean-field dynamics accurately reproduce the many-body solution. Quantum mechanical test calculations for two-state models using

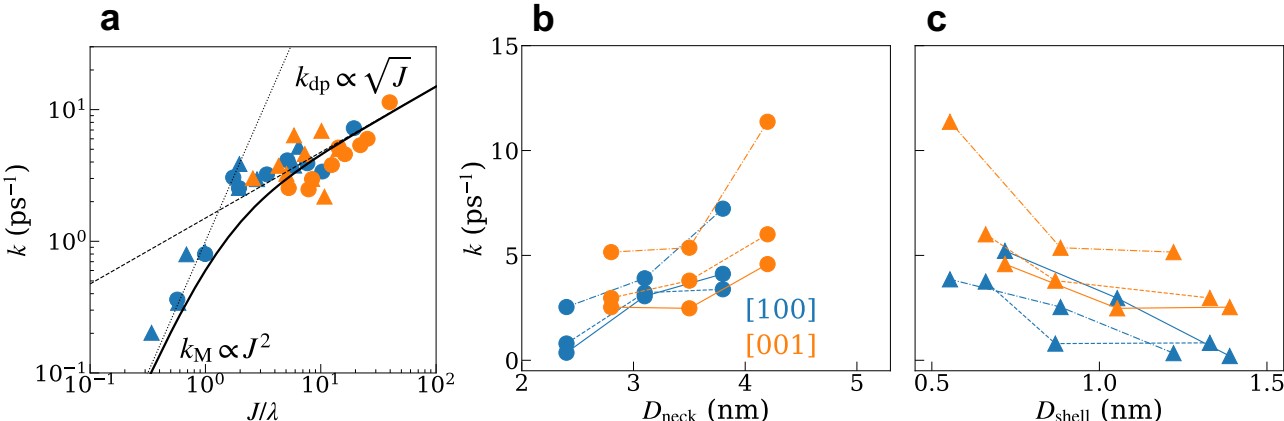

**Fig. 5 | Crossover between different electron transfer regimes. a** Electron transfer rates, $k$, plotted as a function of ground state hybridization/reorganization energy ($J/\lambda$) at 300 K for structures with [100] and [001] attachments. Circle and triangle represent controlling the QD dimer by $D_{neck}$ and $D_{shell}$, respectively. The solid black curve is a fit to a connection formula, $k^{-1} = k_M^{-1} + k_{dp}^{-1}$ with an empirical form for dephasing rate $k_{dp} \propto \sqrt{J}$ shown as a dashed line, and Marcus limit rate $k_M \propto J^2$ shown as a dotted line. **b, c** Electron transfer rates plotted as a function of $D_{neck}$ (**b**) and $D_{shell}$ (**c**) for [100] and [001] orientation attachments. Solid, dashed, and dotted-dashed lines correspond to core diameters $D_{core}$ of 2.2 nm, 3.0 nm, and 3.9 nm. Source data are provided as a Source Data file.

the multi-configuration time-dependent Hartree (MCTDH) method show good agreement with the results of the Ehrenfest method (See Supplementary Figure 8).

We also find that the dephasing rate, physically resulting from coupling to the nuclear vibration, increases with larger hybridization energy, $J$, deep in the adiabatic limit ($\gamma \gg 1$). This seems to be opposite to the behavior expected for the spin-boson model[33], and can be traced to the inclusion of off-diagonal electron-phonon couplings terms in our model Hamiltonian ($V_{n \neq m}^{\alpha}$). As shown in the Supplementary Figure 9 and Fig. 10, the behavior of the population dynamics and the dephasing rates are consistent with the spin-boson model when the off-diagonal coupling terms are turned off ($V_{n \neq m}^{\alpha} = 0$). In addition, the presence of several donor and several acceptor states also affects the dephasing rates, particularly for $\gamma \gg 1$ where the $1P_e$-like donor/acceptor orbitals play a significant role in the charge transfer dynamics due to a smaller energy gap between $1S_e$ and $1P_e$-like states.

To further analyze the dephasing dynamics and delineate the modes that most strongly affect the electron transfer rates, we define the spectral density characterizing the electron-phonon interaction between states $|\phi_n\rangle$ and $|\phi_m\rangle$ ($n,m \in \mathcal{D},A$) as

$$S_{nm}(\omega) = \pi \sum_{\alpha} \omega_{\alpha} \lambda_{nm}^{\alpha} \delta(\omega - \omega_{\alpha}) \tag{7}$$

where $\lambda_{nm}^{\alpha} = \frac{1}{2}\left(\frac{V_{nm}^{\alpha}}{\omega_{\alpha}}\right)^2$ is the reorganization energy for mode $\alpha$. In Fig. 4a, b we plot the diagonal and off-diagonal spectral densities between the ground states of the donor and acceptor ($|\phi_{D_1}\rangle$ and $|\phi_{A_1}\rangle$) for the same systems shown in Fig. 3. The spectral densities are very structured with stronger coupling to the low-frequency acoustic modes ($\omega/2\pi < 1.5$ THz) and weaker coupling to the high-frequency optical modes ($\omega/2\pi > 4$ THz). We find that the overall magnitude of the diagonal spectral densities (Fig. 4a) decreases with increasing $\gamma$ while the off-diagonal spectral densities behave the opposite. Thus, setting the non-diagonal coupling terms to zero leads to a decrease in the dephasing rates with increasing hybridization energies, which is not the case observed in Fig. 3, where the dephasing times are governed by the off-diagonal couplings. In addition, we identify the modes that contribute the most to the dephasing dynamics shown in Fig. 4c, d. The lower frequency mode, $\omega_1$, involves a breathing motion while the higher frequency mode, $\omega_2$ involves a torsional motion (see Supplementary Movie 1 and 2 for details). These vibrational modes are important to localize the charge and facilitate the transfer dynamics.

## Nonadiabatic to adiabatic electron transfer transition

In Fig. 5 we provide a summary of the electron transfer rate constants across the nonadiabatic to adiabatic transition and show how they depend on the neck, core and shell dimensions. In the adiabatic regime, we estimate the electron transfer rate constant by fitting the decay of the envelope of the donor population (see Eq. (20) in Methods). Figure 5a shows a scattered plot of the rate constants calculated for all the dimers considered in this work as a function of the ground state hybridization energy, $J$, at room temperature. As can be seen, the rate constants follow the $J^2$-Marcus theory dependence as $J \to 0$. For larger values of $J$, the population dynamics are no longer characterized by an over-damped exponential decay (unlike the Zusman limit), and the electron transfer is dominated by the dephasing, which shows a weaker dependence on $J$. The first-order dependence of the dephasing rate on $J$ can be derived for the simple two-level spin-boson model, and is given by $k_{dp} = 2S(2J)\coth(J/k_B T)$[34], where $S(\omega)$ is the spectral density defined above. For the multi-state model considered in this work, the dephasing rate depends on other factors discussed previously, such as the off-diagonal spectral densities and contributions from higher-lying states, and increases with $J$ rather than decreases for the standard two-state model. The solid black line in Fig. 5a is a fit to a connection formula for the total electron transfer rate, given by $k^{-1} = k_M^{-1} + k_{dp}^{-1}$ with an empirical form for dephasing rate $k_{dp} \propto \sqrt{J}$. Figure 5b, c show the dependence of $k$ on the neck width $D_{neck}$ and shell thickness $D_{shell}$, for different core diameters $D_{core}$. We find that across the range of neck and shell dimensions that can be varied experimentally[13], the electron transfer rate constants can be tuned over a broad range of values, from approximately 10 ps$^{-1}$ to 100 fs$^{-1}$. This suggests that faster, room-temperature, electron transfer devices require thinner shells, wider necks, and larger cores.

In conclusion, we outlined a theoretical framework to calculate charge transfer between two colloidal quantum dots connected by a bridge. Our approach is based on an atomistic model to describe the electronic structure, the vibrational modes of the donor/acceptor system, the electron-phonon couplings, and used a mixed quantum-classical mean-field Ehrenfest method to describe the time evolution of the reduced density matrix. We showed that by increasing the neck width, increasing the core diameter, and/or decreasing the shell thickness of the quantum dot building blocks, it is possible to control the hybridization energy and tune the system from the Marcus non-adiabatic, slow electron transfer regime to a coherent, adiabatic limit, even at elevated temperatures, with electron transfer times that are orders of magnitude faster and scale mildly with the hybridization

energy. In all regimes, the charge transfer dynamics are mainly governed by the coupling of the electron to inter-dot and torsional acoustic modes.

## Methods

### Nanostructure configurations

The core-shell colloidal quantum dot (CQD) nanocrystals (NCs) were constructed by adding CdS shells to a CdSe core, which was cleaved from a large crystal with a lattice constant of bulk wurtzite CdSe ($a = 4.3$ Å, $c = \sqrt{\frac{8}{3}}a$). The CQD dimers were then constructed by attaching two NCs either through the [100] symmetric or [001] asymmetric crystal plane. The neck/bridge of the dimer can then be widened by adding additional CdS layers to the connection area between two NCs. Supplementary Tables 1 and 2 summarize the different combinations of $D_{neck}$, $D_{shell}$ and $D_{core}$ in two orientations used in this study. The structures were optimized with Stillinger-Weber force field parameterized for II–VI nanostructures[26] using the conjugate gradient minimization implemented in LAMMPS[35]. See Supplementary Data 1 for the relaxed structures.

### Quasi-electron states calculations and localization

The semi-empirical pseudopotential model[20,21] was used to describe the quasi-electron Hamiltonian and the filter-diagonalization technique[22,23] was applied to calculate the eigenstates of the dimer near the bottom of the conduction band. The local screened strain-dependent pseudopotentials following Wang et al.[36] were chosen as the functional form in the momentum space

$$\widetilde{\nu}(q) = a_0 \left[ 1 + a_4 \operatorname{Tr} \boldsymbol{\epsilon} \right] \frac{q^2 - a_1}{a_2 \exp(a_3 q^2) - 1} \tag{8}$$

where $q$ is the momentum, $\boldsymbol{\epsilon}$ is the strain tensor, and the parameters for Cd, Se, and S are collected in Supplementary Table 3. All parameters were fitted to reproduce bulk band structures, band gaps, and effective masses of CdSe and CdS. The real-space quasi-electron Hamiltonian $\hat{h}_{QP}(\mathbf{r})$ is given by

$$\hat{h}_{QP}(\mathbf{r}) = -\frac{1}{2} \nabla_{\mathbf{r}}^2 + \sum_\mu \nu_\mu \left( |\mathbf{r} - \mathbf{R}_{0,\mu}| \right), \tag{9}$$

where $\nu_\mu$ is the real-space pseudopotential for atom $\mu$. The filter-diagonalization technique was then applied to obtain quasi-electron eigenstates, $\psi_i(\mathbf{r})$, above the conduction band edge. The calculations were implemented on real-space grids less than 0.8 a.u. such that the eigenenergies converges less than $10^{-3}$ meV.

To transform the delocalized quasi-electron eigenstates $\psi_i(\mathbf{r})$ to localized donor and acceptor states $\phi_n(\mathbf{r})$, the Förster-Boys localization scheme was applied to maximize the self-extension criteria[24,25]

$$\langle \hat{\Omega} \rangle_{FB} = \sum_{n \in \mathcal{D}, \mathcal{A}} \left( \int d^3 \mathbf{r} |\phi_n(\mathbf{r})|^2 \mathbf{r} \right)^2, \tag{10}$$

where the optimization can be achieved by successive $2 \times 2$ rotations of wavefunction pairs. The resulting localized states are related to eigenstates by a unitary matrix $U$

$$\phi_n(\mathbf{r}) = \sum_i U_{ni} \psi_i(\mathbf{r}). \tag{11}$$

The first 8 eigenstates at the bottom of the conduction band were included to construct localized states.

### Normal modes and vibronic couplings

The nuclear vibration was described by the same 3-body Stillinger-Weber potential used in structure minimization. If the harmonic approximation is made, the normal mode coordinates can be calculated by diagonalizing the Hessian matrix at the equilibrium configuration. For the mass-weighted coordinates, the Hessian matrix is defined as

$$D_{\mu k, \mu' k'} = \frac{1}{\sqrt{m_\mu m_{\mu'}}} \frac{\partial^2 U_{SW}}{\partial u_{\mu k} \partial u_{\mu' k'}} \bigg|_{u=0}, \tag{12}$$

where the atomic displacement $u_{\mu k} = R_{\mu k} - R_{0,\mu k}$, $(k = x, y, z)$. The electron-phonon couplings in the atomic coordinates $V_{nm}^{\mu k}$ are given by the first-order derivative of pseudopotential with respect to the nuclear coordinates[19] and can be transformed to couplings to normal mode $V_{nm}^\alpha$ according to

$$V_{nm}^{\mu k} = \int d\mathbf{r} \phi_n^*(\mathbf{r}) \frac{\partial \nu_\mu \left( |\mathbf{r} - \mathbf{R}_\mu| \right)}{\partial R_{\mu k}} \phi_n(\mathbf{r}) \tag{13}$$

$$V_{nm}^\alpha = \sum_{\mu, k} \frac{1}{\sqrt{m_\mu}} E_{\mu k, \alpha} V_{nm}^{\mu k}, \tag{14}$$

where $E_{\mu k, \alpha}$ are the coefficients of the normal mode transformation.

### Mixed quantum-classical Ehrenfest dynamics

The dynamics of charge transfer were described by the mixed quantum-classical mean-field Ehrenfest method. The system dynamics can be derived from quantum-classical Liouville equation under the mean-field approximation[37,38]

$$\frac{\partial \rho_S(t)}{\partial t} = -\frac{i}{\hbar} \left[ H_S + \sum_\alpha \sum_{n,m \in \mathcal{D},\mathcal{A}} |\phi_n\rangle\langle\phi_m| V_{nm}^\alpha Q_\alpha(t), \rho_S(t) \right], \tag{15}$$

where $\rho_S(t)$ is the reduced density matrix of the electronic system. Under the harmonic approximation, the time evolution of phonon coordinates is

$$\frac{dQ_\alpha(t)}{dt} = P_\alpha(t) \tag{16}$$

$$\frac{dP_\alpha(t)}{dt} = -\omega_\alpha^2 Q_\alpha - \sum_{n,m \in \mathcal{D},\mathcal{A}} V_{nm}^\alpha \operatorname{Tr}_S \{ |\phi_n\rangle\langle\phi_m| \rho_S(t) \}. \tag{17}$$

The above equations of motion in ((15)–(17)) can be integrated by the fourth order Runge-Kutta method with a time step around 1 fs. The phonon coordinates are integrated taking $\rho_S(t)$ to be constant over a half-time step. The initial density matrix was assumed to be separated into a product of the equilibrated system and bath density matrices

$$\rho(0) = \rho_S(0)\rho_B(0) = \left( \sum_{n \in D} \frac{e^{-\beta E_n}}{Z_D} |n\rangle\langle n| \right) \frac{e^{-\beta H_B}}{Z_B}, \tag{18}$$

where $Z_D$ and $Z_B$ are partition functions of the donor and phonon subspaces, respectively. The initial phonon coordinates $P_\alpha$ and $Q_\alpha$ are sampled from the Boltzmann distribution and the populations $\rho_S(t)$ are averaged over the ensemble trajectories. The calculations required an average of 1000–4000 trajectories to converge. The anharmonicity effects can be included by propagating the atomic coordinates in the force field of Stillinger-Weber potential. We found that the anharmonic nuclear coordinate does not change the dynamics significantly.

The population transfer between donor and acceptor subspaces can be described by $p_\mathcal{D}(t)$ and $p_\mathcal{A}(t)$, which are obtained by summing over the populations in the donor and acceptor Hilbert space,

respectively

$$p_{\mathcal{D}/\mathcal{A}}(t) = \sum_{n \in \mathcal{D}/\mathcal{A}} \rho_{S,nn}(t). \tag{19}$$

In general, there are two important time scales associated with charge transfer dynamics, and it is possible to fit an envelop function $p_{\mathrm{env}}(t)$ of $p_{\mathcal{D}}(t)$ (or $p_{\mathcal{A}}(t)$) as a sum of exponential functions:

$$p_{\mathrm{env}}(t) = A_{\mathrm{M}} e^{-k_{\mathrm{M}}t} + A_{\mathrm{dp}} e^{-k_{\mathrm{dp}}t}, \tag{20}$$

where $k_{\mathrm{M}}$ is Marcus rate, $k_{\mathrm{dp}}$ is the dephasing rate, and $A$ parameters are weighted factors. We chose $A_{\mathrm{dp}} = 0$ in the nonadiabatic Marcus regime and $A_{\mathrm{M}} = 0$ in the adiabatic coherent regime. The total transfer rate $k$ is chosen as either $k_{\mathrm{M}}$ in the nonadiabatic regime or $k_{\mathrm{dp}}$ in the adiabatic regime.

## Data availability
The data supporting the findings of this study are available within the article and Supplementary Information. Source data are provided with this paper. Extra data are available upon request. Source data are provided with this paper.

## Code availability
Computer codes for all simulations and analysis in this paper are available at: https://doi.org/10.5281/zenodo.7686521.

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

## Acknowledgements
This work was supported by the NSF-BSF International Collaboration in the Division of Materials Research program, NSF grant number DMR-2026741 and BSF grant number 2020618. Methods used in this work were provided by the Center for Computational Study of Excited State Phenomena in Energy Materials (C2SEPEM), which is funded by the U.S. Department of Energy, Office of Science, Basic Energy Sciences, Materials Sciences and Engineering Division, via contract no. DE-AC02-05CH11231, as part of the Computational Materials Sciences Program. Computational resources were provided in part by the National Energy Research Scientific Computing Center (NERSC), a U.S. Department of Energy Office of Science User Facility operated under contract no. DE-AC02- 05CH11231. We thank Dipti Jasrasaria and Daniel Weinberg for useful discussions and helping calculating the vibronic couplings.

## Author contributions
B.H., U.B., and E.R. designed the research and conceived the project. B.H. and E.R. contributed to the development of the computational model and produced the results. B.H. and M.T. validated the computational framework. All authors were involved in discussing the result and writing the manuscript and Supplementary Information.

## Competing interests
The authors declare no competing interests.
