## [Peer Review File · Nature Communications]

Incoherent Nonadiabatic to Coherent Adiabatic Transition of Electron Transfer in Colloidal Quantum Dot MoleculesReviewers' Comments:

Reviewer #1:

Remarks to the Author:

In this work, Rabani and coworkers report on a theoretical framework to analyze different regimes of charge transfer between two colloidal quantum dots connected by a bridge. By varying the characteristic parameters of the quantum dot building blocks – neck size, core size, or shell thickness – the authors show that their atomistic model can reproduce different regimes of electron transfer, from Marcus nonadiabatic transfer to the adiabatic limit.

This work presents an impressive set of charge-transfer atomistic simulations of colloidal quantum dots that unravel the passage through different electron-transfer regimes. While I believe that this work could be of interest to the readership of Nature Communications, I have a few concerns that the authors may need to address.

My first concern is linked to the lack of validation of the theoretical formalism to reproduce experimental observables related to the colloidal quantum dots. If the goal of this work is to stimulate new experiments on this system, I would have perhaps expected that the authors find some observables to benchmark and validate their theoretical framework, bringing more confidence in the predictions discussed here. Ref. [12] reports, for example, absorption spectra. Would it be possible for the authors to provide further validation of their theoretical formalism?

Connected to this first issue, the authors use Ehrenfest dynamics for the three studied regimes. Could the authors possibly provide a validation for using this mixed quantum/classical method for the different types of charge-transfer mechanisms under investigation? Any shortcomings emerging from its mean-field nature? The authors used a time step of 1 fs for the phonon dynamics, but what was the time step associated with the integration of Eq. (12)?

Reviewer #2:

Remarks to the Author:

The manuscript "Nonadiabatic to Adiabatic Transition of Electron Transfer in Colloidal Quantum Dot Molecules" by Huo et al. is a theoretical/numerical study on the quantum dynamics in a colloidal quantum dot molecule. Using a microscopic theory, the parameters for the electron transfer are derived which results in different dynamics based on different tuning knobs like core/shell size and neck thickness. The result is that a great variety in dynamics can be achieved in such systems.

The manuscript is well written and is well prepared. The results seem to be sound (as far as one can check this without repeating the simulations).

My main problem with the manuscript is that I did not get excited about the results at all. My background is not in electronic transfer, but I work on quantum dots and optical passages, for which I see a lot of similarities to what is found in this study. Hence, I was able to follow the manuscript easily. But I could neither see the big novelty in the manuscript. The paper strongly uses the community language, doesn't explain what is meant by adiabatic/non-adiabatic or what the larger picture is.

Therefore, I do not think that this paper is a good fit to Nature Communication, but could be well published in a more specialized journal.

Reviewer #3:

Remarks to the Author:

The manuscript by Hou et al. reports on a theoretical investigation of nonadiabatic to adiabatic transition of electron Transfer in colloidal quantum dot molecules. The authors investigate a colloidal

quantum dot dimer with thousands of atoms by combining Mixed quantum-classical Ehrenfest dynamics and a semi-empirical pseudopotential model. The highlight quantum dot dimer behaves from the adiabatic to the nonadiabatic limit of electron transfer by turning the neck and core sizes and decreasing the shell thickness.

The topic of charge transfer in this article is interesting. Using a series of Quantum Dot Molecules to probe different electron transfer regimes is an exciting challenge. However, I have a few general issues with the manuscript in its current form that prevent me from recommending it for publication.

1. Zhu et.al have reported a realization of the adiabatic and nonadiabatic electron transfer by changing the bridge length connecting the donor and acceptor and the functional groups in 2021. The manuscript has limited novelty.
2. In figure 2c, the authors found an excellent correlation between J/λ and the adiabatic parameter. J is the coupling energy between the donor and acceptor states, and λ is the reorganization energy. With the physical picture of adiabatic and diabatic potential energy surfaces in figure 1, the reasonable range of J/λ should be smaller than 0.5. Please clarify the range of J/λ .
3. Please clarify the physical source of the dephasing of electron transfer.
4. I suggest marking the donor and acceptor state in figure 1b and what the color represents of the wavefunction in figure b.

Response to Reviewer 1.

Comment 1.

In this work, Rabani and coworkers report on a theoretical framework to analyze different regimes of charge transfer between two colloidal quantum dots connected by a bridge. By varying the characteristic parameters of the quantum dot building blocks - neck size, core size, or shell thickness - the authors show that their atomistic model can reproduce different regimes of electron transfer, from Marcus nonadiabatic transfer to the adiabatic limit. This work presents an impressive set of charge-transfer atomistic simulations of colloidal quantum dots that unravel the passage through different electron-transfer regimes. While I believe that this work could be of interest to the readership of Nature Communications, I have a few concerns that the authors may need to address.

Reply 1.

We thank the reviewer for the positive assessment of our work and for finding it interesting for the readership of Nature Communications.

Comment 2.

My first concern is linked to the lack of validation of the theoretical formalism to reproduce experimental observables related to the colloidal quantum dots. If the goal of this work is to stimulate new experiments on this system, I would have perhaps expected that the authors find some observables to benchmark and validate their theoretical framework, bringing more confidence in the predictions discussed here. Ref. [12] reports, for example, absorption spectra. Would it be possible for the authors to provide further validation of their theoretical formalism?

Reply 2.

We thank the reviewer for the comment. The reviewer concerns about the validation of our theoretical formalism related to the experimental observables in colloidal quantum dots. In fact, we have validated our theoretical models on both electronic, excitonic and vibronic properties of the colloidal quantum dot monomers and dimers in previous work. For examples:

(1) Using the semiempirical pseudopotential method combined with the Bethe–Salpeter equation (BSE) [1] in the static screening approximation [2] we have calculated the optical gap and compared our predictions with experimental measurements for several different NC compositions and size, as illustrated in Fig. 1. We find very good agreement between the computed and measured gaps across all systems considered, validating our pseudopotential model and the approach.

FIG. 1: Figure adopted from Ref. [3]. Gaps for wurtzite CdSe quantum dots of different sizes (left). The optical gaps computed by our semiempirical pseudopotential method agree with experimental measurements of the optical gap by Fan *et al.* [4] (black squares) and Yu *et al.* [5] (black triangles). The inset shows the exciton binding energy, E_B , computed by our method and compared to the values computed by Franceschetti and Zunger [6] (black asterisks). Gaps for zincblende CdSe–CdS core–shell nanoplatelets with different thicknesses of the CdS shell (middle). The optical gaps calculated by our method compare favorably with those measured experimentally by Hazarika *et al.* [7] (black squares). Gaps for zincblende InAs quantum dots of different sizes (right). The fundamental gaps calculated are in excellent agreement with those measured by Banin *et al.* [8] using scanning tunneling microscopy (black squares), and the optical gaps compare well with those measured by Guzelian *et al.* [9] (black triangles) and computed by Franceschetti and Zunger [6] (black asterisks).

(2) The reorganization energies (correlate with the experimentally measured Stokes shifts) for wurtzite CdSe calculated using exciton-phonon coupling obtained from the pseudopotential model are shown in Fig. 2). We find that with no adjustable parameters, the calculated values of the reorganization energy compare favorably with values obtained from a parametrized effective mass model and from room-temperature absorption and photoluminescence measurements of the Stokes shift.

(3) The photoluminescence (PL) spectra of a 3nm CdSe core/3 monolayer (ML) CdS shell nanocrystal (NC) at different temperatures calculated using the pseudopotential model combined with linear response theory, is shown in Fig. 3 (black and red curves for first and second

FIG. 2: Figure adopted from Ref. [10]. Reorganization energies of exciton for CdSe NCs of various diameters in comparison to values from effective mass model-based calculations by Kelley [11] and from experimental measurements by Bawendi *et al.* [12] and Scholes *et al.* [13] (left). Acoustic modes contribute more significantly to the reorganization energy than optical modes (right).

order coupling models), and compared to single NC PL measurements (green curves). Our calculations are in very good agreement with the single-NC PL measurements at low and intermediate temperatures, providing a quantitatively accurate description of the relative positions and intensities of the zero-phonon line (ZPL) and the phonon sidebands, as well as their temperature dependence. At higher temperatures, the inclusion of the second-order exciton-phonon coupling is required to account for high temperature lattice fluctuations.

(4) We have also calculated the nonradiative Auger recombination (AR) rates using the pseudopotential model with the BSE and Fermi's golden rule, and compared our prediction

FIG. 3: Figure adopted from Ref. [14]. Single-molecule photoluminescence spectra for a 3nm diameter CdSe / 3 monolayer CdS core-shell nanocrystal at temperatures ranging from 4 to 290 K. The calculated results from the model Hamiltonian (black curves) and from the empirical inclusion of second-order expansion of exciton-phonon couplings (red curves) are compared to the experimental measurements (green curves).

FIG. 4: Figure adopted from [15]. AR lifetimes, τ_{AR} , for CdSe QDs as a function of the volume of the QD. Good agreement is observed between the interacting formalism (green circles) and experimental (blue squares: solid [16], vertical lines [17], and horizontal lines [18]) AR lifetimes for all sizes. On the other hand, the noninteracting formalism (red triangles) deviates from the experimental values for QD volumes $>10 \text{ nm}^3$. Power law fits, $\tau_{AR} = a \times V_b$, are also shown for each of the three sets of AR lifetimes.

with experimental measurements of the AR lifetimes,[16–18] as shown in Fig. 4. The overall agreement with the experiments is remarkable as well as the first approach to capture the ”universal” volume scaling of AR lifetimes.

(5) In Fig. 5(a) we compare the optical shift between the QD monomers and the corresponding QD dimers, using the semiempirical pseudopotential method combined with the BSE. We find an excellent agreement between the measured (red stars) and the calculated (green squares) shifts when comparing similar shell thicknesses ($R_S = 2.1\text{nm}$). Our model allows delineating the role of strain, deconfinement, and hybridization.

FIG. 5: Figure adopted from [19]. (a) The shift in the optical gaps, δE_X , and (b) the shift in the exciton binding energy, E_B , as function of neck width for QDs with a total diameter $D = 4.6$ nm (blue symbols), $D = 5.6$ nm (orange symbols), and $D = 7$ nm (green symbols). Circles (squares) connected by solid (dashed) line correspond to the [100] ([001]) plane of attachment. Experimental shifts in the optical gaps of $D = 7$ nm QDs (red asterisks) are included for comparison [20]. (c) The shift of the optical gaps (orange symbols), for the $D = 5.6$ nm series of dimers compared to Eq. (2) (red crosses).

In summary, previous validations of our approach indicate that it is reliable in predicting the exciton optical gap, reorganization energy/Stokes shift, photoluminescence spectral line shape, Auger recombination rate, and exciton hybridization energies in QD dimer. In order to prevent confusion, we have added citations to the above validations, and added the following sentence on page 3, "The approach was recently validated in predicting optical gap, reorganization energies/Stokes shifts, linear optical PL spectrum, multi-excitonic effects, and more [14-18]."

Comment 3.

Connected to this first issue, the authors use Ehrenfest dynamics for the three studied regimes. Could the authors possibly provide a validation for using this mixed quantum/classical method for the different types of charge-transfer mechanisms under investigation? Any shortcomings emerging from its mean-field nature?

Reply 3.

We thank the reviewer for the comment. The performance of Ehrenfest dynamics can be validated by comparing it with numerically exact methods such as the multi-configurational time-dependent Hartree (MCTDH) approach [21–24]. The MCTDH calculations done here for validation were performed with the Heidelberg MCTDH package [25]. Fig. 6 shows the comparison of population dynamics calculated from Ehrenfest and MCTDH approach in nonadiabatic ($\gamma \ll 1$), intermediate ($\gamma \approx 1$), and adiabatic ($\gamma \gg 1$) regimes with ground donor and acceptor states. The two methods agree very well across all electron transfer regimes in the parameter space we considered. In the nonadiabatic regime, Ehrenfest and MCTDH also agree with the exponential decay with transition rates obtained from Marcus theory shown in Fig. 6(a). Fig. 6(c) validates the adiabatic dynamics in asymmetrically attached QD dimers where an energy bias $\Delta\varepsilon$ is present.

In addition, we provide a technical argument in the main text to why the mean-field approach works well in all three regimes studied here: **”The agreement between the mean-field theory and the perturbative Marcus result for $\gamma \ll 1$ is consistent with the so-called average classical limit, where the dynamics of the nuclear degrees of freedom are governed by the arithmetic average donor/acceptor Hamiltonian... For large values of γ , where the hybridization energy is much larger than the reorganization energy ($J \gg \lambda$) and the system is in the weak electron-phonon coupling limit, the average mean force on the nuclei is similar to the force for each diabatic potential energy surfaces, and the mean-field dynamics accurately reproduce the many-body solution...”**

FIG. 6: Comparison of the population dynamics calculated by Ehrenfest and MCTDH methods across different regimes. Blue and red lines represent donor and acceptor populations. (a)-(c) ground-state two-level dynamics are extracted from the multi-level dynamics of (1a), (1e) and (2b) respectively shown in the Supporting information Figure S3. and Figure S4.

Comment 4.

The authors used a time step of 1 fs for the phonon dynamics, but what was the time step associated with the integration of Eq. (12)?

Reply 4.

For the step of integration, we use $\Delta t = 1$ fs for integrating both system and phonon dynamics. We use 4th order Runge–Kutta methods to integrate the equation of motion, whose local truncation error scales as $\mathcal{O}(\Delta t^5)$ (convergence plot of Δt is shown in Fig 7). Note that the bath equation of motion is integrated numerically taking $\rho_S(t)$ to be constant over a half-time step

$$\begin{aligned}
 Q_\alpha(t + \frac{\Delta t}{2}) &= \left[Q_\alpha(t) + \frac{1}{\omega_\alpha^2} \sum_{n,m \in \mathcal{D}, \mathcal{A}} V_{nm}^\alpha \text{Tr}_S \{ |\phi_n\rangle \langle \phi_m| \rho_S(t) \} \right] \cos\left(\omega_\alpha \frac{\Delta t}{2}\right) + \frac{P_\alpha(t)}{\omega_\alpha} \sin\left(\omega_\alpha \frac{\Delta t}{2}\right) \\
 &\quad - \frac{1}{\omega_\alpha^2} \sum_{n,m \in \mathcal{D}, \mathcal{A}} V_{nn}^\alpha \text{Tr}_S \{ |\phi_n\rangle \langle \phi_n| \rho_S(t) \} \\
 P_\alpha(t + \frac{\Delta t}{2}) &= - \left[Q_\alpha(t) + \frac{1}{\omega_\alpha^2} \sum_{n,m \in \mathcal{D}, \mathcal{A}} V_{nm}^\alpha \text{Tr}_S \{ |\phi_n\rangle \langle \phi_m| \rho_S(t) \} \right] \omega_\alpha \sin\left(\omega_\alpha \frac{\Delta t}{2}\right) + \frac{P_\alpha(t)}{\omega_\alpha} \cos\left(\omega_\alpha \frac{\Delta t}{2}\right).
 \end{aligned} \tag{1}$$

We rewrite our description in page 10 as ”The above equations of motion in (12-14) can be integrated by the fourth order Runge-Kutta method with a time step around 1 fs. The phonon coordinates are integrated taking $\rho_S(t)$ to be constant over a half-time step”.

FIG. 7: Convergence of integration time step Δt for Ehrenfest dynamics. The dots represent the relative error with respect to a single population trajectory with respect to $\Delta t = 0.25$ fs. The population dynamics is well convergent with $\Delta t = 1$ fs.

Response to Reviewer 2.

Comment 1.

The manuscript “Nonadiabatic to Adiabatic Transition of Electron Transfer in Colloidal Quantum Dot Molecules” by Huo et al. is a theoretical/numerical study on the quantum dynamics in a colloidal quantum dot molecule. Using a microscopic theory, the parameters for the electron transfer are derived which results in different dynamics based on different tuning knobs like core/shell size and neck thickness. The result is that a great variety in dynamics can be achieved in such systems.

The manuscript is well written and is well prepared. The results seem to be sound (as far as one can check this without repeating the simulations).

Reply 1.

We thank the reviewer for the in-depth review and support for our results. In the following, we will address the comments one by one.

Comment 2.

My main problem with the manuscript is that I did not get excited about the results at all. My background is not in electronic transfer, but I work on quantum dots and optical passages, for which I see a lot of similarities to what is found in this study. Hence, I was able to follow the manuscript easily. But I could neither see the big novelty in the manuscript.

Reply 2.

We thank the reviewer for the valuable comments and for being able to follow our manuscript, despite being outside the community. Electron transfer is indeed an “old” problem. Most of our intuition and understanding of electron transfer relies on intra and intermolecular electron transfer in solution on one hand and molecular junctions, molecule/semiconductor, and semiconductor/semiconductor interfaces on the other hand. Such control is required in order to drive the mechanism of electron transfer and achieve ultrafast charge transfer devices. Indeed, only very recently it was demonstrated that such a transition can be systematically tuned in a multi-valence donor-bridge-acceptor complex [26]. In comparison, our system offers more controls over the donor and acceptor couplings, in a regime that results in ultrafast coherent electron transfer at room tempera-

tures, rather than the over-damped molecular limit [26]. Coherence at room temperature is also important for other quantum information-based applications. We believe that while our work is computational and predictive, our model is well tested (see reply to Referee 1), and the novel results will generate interest in the community in both confirming our predictions as well as in searching for optimal charge transfer devices based on quantum dot dimers. We also added "The flexibility of controlling the hybridization energy without affecting the other energy scales in the system, offers a platform to drive the system from the over-damped electron transfer dynamics typical to molecular systems to the coherent limit, where the electron transfer rates are governed by decoherence times, even at elevated temperatures." on page 3 to emphasize our novelty.

Comment 3.

The paper strongly uses the community language, doesn't explain what is meant by adiabatic/non-adiabatic or what the larger picture is.

Reply 3.

We thank the reviewer for the comment. We have better explained what is meant by non-adiabatic in the main text by modifying the introduction into "The theoretical framework to describe charge transfer reactions in condensed phases dates back to the seminal work of Marcus [1], where he considered a donor molecule *weakly* coupled to an acceptor molecule, and developed a theoretical framework to describe the electron transfer reactions in a fluctuating environment." and "This nonadiabatic limit is characterized by a small value of the so-called "adiabatic parameter", γ , defined as the ratio between the donor-acceptor hybridization (J), the characteristic nuclear vibrational (ω_c), and the reorganization energy λ (see Fig. 1(a) for a sketch of these energy parameters) [2]:

$$\gamma = \frac{|J|^2}{2\hbar\omega_c} \sqrt{\frac{\pi}{\lambda k_B T}}, \quad (2)$$

The nonadiabatic electron transfer Marcus regime is thus achieved either for weak electronic coupling and/or for fast nuclear motion ($\omega_c > J/\hbar$).", when we discuss Fig.1, on page 2. We also add "Zusman developed a framework to describe the crossover from the Marcus weak coupling nonadiabatic limit to the *adiabatic* limit [7,8], where the coupling between donor and acceptor is large and/or the nuclei are slow ($\gamma \gg 1$). In the adiabatic limit,

diabatic states are no longer a good representation and the dynamics proceed on a single adiabatic (Born-Oppenheimer) surface (see solid lines in Fig. 1(a)).” when discussing adiabatic electron transfer on page 3.

Comment 4.

Therefore, I do not think that this paper is a good fit to Nature Communication, but could be well published in a more specialized journal.

Reply 4.

We thank the reviewer for providing valuable comments and suggestions, which helped us improve the manuscript. We hope that our reply to the referee’s comments and to the comments of the other referees changes the reviewer’s mind about our work. We feel, as do the other two referees, that the work (with proper adjustments) does present work suitable for publication in *Nature Communications*.

Response to Reviewer 3.

Comment 1.

The manuscript by Hou et al. reports on a theoretical investigation of nonadiabatic to adiabatic transition of electron Transfer in colloidal quantum dot molecules. The authors investigate a colloidal quantum dot dimer with thousands of atoms by combining Mixed quantum-classical Ehrenfest dynamics and a semi-empirical pseudopotential model. The highlight quantum dot dimer behaves from the adiabatic to the nonadiabatic limit of electron transfer by turning the neck and core sizes and decreasing the shell thickness.

The topic of charge transfer in this article is interesting. Using a series of Quantum Dot Molecules to probe different electron transfer regimes is an exciting challenge. However, I have a few general issues with the manuscript in its current form that prevent me from recommending it for publication.

Reply 1.

We thank the reviewer for the in-depth review and high evaluation of our work. In the following, we will address the comments one by one.

Comment 2.

1. Zhu et.al have reported a realization of the adiabatic and nonadiabatic electron transfer by changing the bridge length connecting the donor and acceptor and the functional groups in 2021. The manuscript has limited novelty.

Reply 2.

Please see our reply to comment 2 of referee 2. In addition, we would like to add the following more technical discussions: our work is different from the multi-valence donor-bridge-acceptor in Zhu *et al.* (cited by us) in two different ways. First, electron transfer in molecular bridges is in the strong coupling limit to the solvent ($0.005 < J/\lambda < 0.3$), resulting in over-damped dynamics in both adiabatic and Marcus limits. The QD dimer, on the other hand, covers much broader dynamical regimes ($0.2 < J/\lambda < 70$), in which both transitions from nonadiabatic to adiabatic and incoherent to coherent electron transfer can be realized. Second, we find that the QD dimers have better tunability over molecular systems since changing the electronic couplings can be achieved by increasing the neck sizes or core-to-shell ratio, and does not significantly affect the reorganization and characteristic vibrational frequency. Therefore, we believe our study provide a novel platform to realize

the diverse behavior of electron transfer.

Comment 3.

2. In figure 2c, the authors found an excellent correlation between J/λ and the adiabatic parameter. J is the coupling energy between the doner and acceptor states, and λ is the reorganization energy. With the physical picture of adiabatic and diabatic potential energy surfaces in figure 1, the reasonable range of J/λ should be smaller than 0.5. Please clarify the range of J/λ .

Reply 3.

We thank the reviewer for the comment. The potential energy surface in Fig.1 (a) (in main text) is just a schematic illustration of the parameters in the *nonadiabatic* Marcus rate expression in Eq. (1). Indeed in this limit, J/λ cannot exceed 0.5 for a barrier to exist as shown in the figure. In general, J/λ is not limited to 0.5 in the intermediate to the adiabatic regime and can be any value as long as it is experimentally achievable. Fig. 8 below shows the potential energy surfaces with $J/\lambda = 1$ for electron transfer in the intermediate/adiabatic regime. In this case, there will be no energy barrier and the dynamics will be coherent. To avoid possible confusion, we add "(a) Schematic sketch of the potential energy surfaces along the reaction coordinate [1] Q for the *nonadiabatic* electron transfer and the energy scales appearing in Eq. (1)." in the caption of Fig. 1(a).

FIG. 8: Schematic sketch of the potential energy surfaces with $J/\lambda = 1$. The solid and dashed lines represent the adiabatic and diabatic potential energy surfaces, respectively.

Comment 4.

3. Please clarify the physical source of the dephasing of electron transfer.

Reply 4.

We thank the reviewer for the comment. As indicated in the manuscript, there are several factors that contribute to the dephasing of electron transfer. We showed in Fig. 4(a) (in the main text) that the spectral density $S(\omega)$ peaks at around 0.6 THz, which means that the electronic system couples most strongly to those low-frequency vibrations. Thus, those low-frequency vibrations contribute to the time scale of dephasing in the population dynamics. Fig. 9 below compares the population dynamics with and without the vibrational modes below the characteristic frequency $\omega_c/2\pi = 0.6$ THz, which shows that the dephasing time becomes much longer when removing those low-frequency modes. We modify the sentence on page 6 to emphasize this "We also find that the dephasing rate, physically resulting from coupling to the nuclear vibration, increases with increasing hybridization energy, J , deep in the adiabatic limit ($\gamma \gg 1$).". We also added a section called "Physical Source of Dephasing" in the supporting information to discuss this point.

Another source of dephasing comes from populating different donor and acceptor states. As shown in Fig. S2(c) in the Supporting Information, the population oscillates between D_1/A_1 ($1S_e$ -like states) and D_2/A_2 ($1P_e$ -like states), with different dephasing rates, controlled by the off-diagonal couplings $V_{n \neq m}^\alpha$ (in main text Fig. 4(b)) to the phonons.

FIG. 9: Comparison of the dynamics with and without low-frequency modes below ω_c . The original dynamics is from Figure S4 (2b) in supporting information.

Comment 5.

4. I suggest marking the donor and acceptor state in figure 1b and what the color represents of the wavefunction in figure b.

Reply 5.

We thank the reviewer for the valuable suggestion. Fig. 1 in the main text has been modified into Fig. 10 below to follow our convention of coloring donor states as blue and acceptor states as red and modify the figure caption on page 2 to ”(b) Hyper-sphere plots of the donor (left NC, blue) and acceptor states (right NC, red) were obtained from the semi-empirical pseudopotential calculation and FB-localization of the quasi-electron eigenstates in wurtzite CQD dimers. The dark and light colors of the wavefunctions indicate positive and negative phases respectively.”. We also modify Fig. S1 in the Supporting Information as Fig. 11.

FIG. 10: Modification of Fig.1 in the main text.

FIG. 11: Modification of Fig.S1 in the Supporting Information.

-
- [1] Strinati, G. Effects of dynamical screening on resonances at inner-shell thresholds in semiconductors. *Phys. Rev. B* **29**, 5718–5726 (1984).
- [2] Rohlfing, M. & Louie, S. G. Electron-hole excitations and optical spectra from first principles. *Phys. Rev. B* **62**, 4927–4944 (2000).
- [3] Jasrasaria, D., Weinberg, D., Philbin, J. P. & Rabani, E. Simulations of nonradiative processes in semiconductor nanocrystals. *J. Chem. Phys.* **157**, 020901 (2022).
- [4] Fan, K. *et al.* Effect of shell thickness on electrochemical property of wurtzite CdSe/CdS core/shell nanocrystals. *Chemical Physics Letters* **633**, 1–5 (2015).
- [5] Yu, W. W., Qu, L., Guo, W. & Peng, X. Experimental Determination of the Extinction Coefficient of CdTe, CdSe, and CdS Nanocrystals. *Chem. Mater.* **15**, 2854–2860 (2003).
- [6] Franceschetti, A. & Zunger, A. Direct Pseudopotential Calculation of Exciton Coulomb and Exchange Energies in Semiconductor Quantum Dots. *Phys. Rev. Lett.* **78**, 915–918 (1997).
- [7] Hazarika, A. *et al.* Colloidal Atomic Layer Deposition with Stationary Reactant Phases Enables Precise Synthesis of “Digital” II–VI Nano-heterostructures with Exquisite Control of Confinement and Strain. *J. Am. Chem. Soc.* **141**, 13487–13496 (2019).
- [8] Banin, U., Cao, Y., Katz, D. & Millo, O. Identification of atomic-like electronic states in indium arsenide nanocrystal quantum dots. *Nature* **400**, 542–544 (1999).
- [9] Guzelian, A. A., Banin, U., Kadavanich, A. V., Peng, X. & Alivisatos, A. P. Colloidal chemical synthesis and characterization of InAs nanocrystal quantum dots. *Appl. Phys. Lett.* **69**, 1432–1434 (1996).
- [10] Jasrasaria, D. & Rabani, E. Correction to Interplay of Surface and Interior Modes in Exciton–Phonon Coupling at the Nanoscale. *Nano Lett.* **22**, 8033–8034 (2022).
- [11] Kelley, A. M. Electron–Phonon Coupling in CdSe Nanocrystals from an Atomistic Phonon Model. *ACS Nano* **5**, 5254–5262 (2011).
- [12] Liptay, T. J., Marshall, L. F., Rao, P. S., Ram, R. J. & Bawendi, M. G. Anomalous Stokes shift in CdSe nanocrystals. *Phys. Rev. B* **76**, 155314 (2007).
- [13] Salvador, M. R., Graham, M. W. & Scholes, G. D. Exciton-phonon coupling and disorder in the excited states of CdSe colloidal quantum dots. *The Journal of Chemical Physics* **125**, 184709 (2006).

- [14] Lin, K. *et al.* Theory of photoluminescence spectral line shapes of semiconductor nanocrystals (2022). ArXiv:2212.06323.
- [15] Philbin, J. P. & Rabani, E. Electron–Hole Correlations Govern Auger Recombination in Nanostructures. *Nano Lett.* **18**, 7889–7895 (2018).
- [16] Klimov, V. I., Mikhailovsky, A. A., McBranch, D. W., Leatherdale, C. A. & Bawendi, M. G. Quantization of Multiparticle Auger Rates in Semiconductor Quantum Dots. *Science* **287**, 1011–1013 (2000).
- [17] Htoon, H., Hollingsworth, J. A., Dickerson, R. & Klimov, V. I. Effect of Zero- to One-Dimensional Transformation on Multiparticle Auger Recombination in Semiconductor Quantum Rods. *Phys. Rev. Lett.* **91**, 227401 (2003).
- [18] Taguchi, S., Saruyama, M., Teranishi, T. & Kanemitsu, Y. Quantized Auger recombination of biexcitons in CdSe nanorods studied by time-resolved photoluminescence and transient-absorption spectroscopy. *Phys. Rev. B* **83**, 155324 (2011).
- [19] Verbitsky, L., Jasrasaria, D., Banin, U. & Rabani, E. Hybridization and deconfinement in colloidal quantum dot molecules. *J. Chem. Phys.* **157**, 134502 (2022).
- [20] Cui, J. *et al.* Neck Barrier Engineering in Quantum Dot Dimer Molecules via Intraparticle Ripening. *J. Am. Chem. Soc.* **143**, 19816–19823 (2021).
- [21] Meyer, H. D., Manthe, U. & Cederbaum, L. S. The multi-configurational time-dependent Hartree approach. *Chemical Physics Letters* **165**, 73–78 (1990).
- [22] Manthe, U., Meyer, H. & Cederbaum, L. S. Wave-packet dynamics within the multiconfiguration Hartree framework: General aspects and application to NOCl. *The Journal of Chemical Physics* **97**, 3199–3213 (1992).
- [23] Beck, M. H., Jäckle, A., Worth, G. A. & Meyer, H. D. The multiconfiguration time-dependent Hartree (MCTDH) method: a highly efficient algorithm for propagating wavepackets. *Physics Reports* **324**, 1–105 (2000).
- [24] Wang, H. & Thoss, M. Multilayer formulation of the multiconfiguration time-dependent Hartree theory. *J. Chem. Phys.* **119**, 1289–1299 (2003).
- [25] Beck, M. H., Jäckle, A., Worth, G. A. & Meyer, H. D. The mctdh package, version 8.5 (2020). URL <http://mctdh.uni-hd.de/>.
- [26] Zhu, G. Y. *et al.* Crossover between the adiabatic and nonadiabatic electron transfer limits in the Landau-Zener model. *Nat. Commun.* **12**, 456 (2021).

Reviewers' Comments:

Reviewer #1:

Remarks to the Author:

I thank the authors for their thorough and convincing answers to my questions. I would support publication of this revised manuscript in Nature Communications.

Reviewer #2:

Remarks to the Author:

This is my second report to the manuscript "Nonadiabatic to Adiabatic Transition of Electron Transfer in Colloidal Quantum Dot Molecules" by Huo et al. Three reviews were performed and the authors have answered to the questions raised.

From my point of view, the paper is technically sound and the authors took care to use valid approaches. As previously mentioned, the paper is well written and easy to follow. Hence, the paper is surely publishable.

The main concern that I raised is that the paper has a rather limited audience and is not accessible to people outside the community. The authors have replied to my comment and added a few remarks to the paper. While these comments have improved the manuscript, I am still not excited about it and still think the value of the work is not easily accessible for a general audience. But I can acknowledge that coherence at room temperature is a very crucial task for quantum technologies and that the research might spark experiments. Thus, it might fit into the scope of Nature Communications.

Reviewer #3:

Remarks to the Author:

In the revised version of "Incoherent Nonadiabatic to Coherent Adiabatic Transition of Electron Transfer in Colloidal Quantum Dot Molecules", the authors provided further explanations to confirm the novelty. And I am satisfied with the author's response to my comments. And recommend the current version could be published in NATURE COMM without further revisions.

Response to Reviewer 1.

Comment

I thank the authors for their thorough and convincing answers to my questions. I would support publication of this revised manuscript in Nature Communications.

Reply

We would like to thank the reviewer for the thoughtful comments and suggestions on our original manuscript.

Response to Reviewer 2.

Comment

This is my second report to the manuscript “Nonadiabatic to Adiabatic Transition of Electron Transfer in Colloidal Quantum Dot Molecules” by Huo et al. Three reviews were performed and the authors have answered to the questions raised.

From my point of view, the paper is technically sound and the authors took care to use valid approaches. As previously mentioned, the paper is well written and easy to follow. Hence, the paper is surely publishable.

The main concern that I raised is that the paper has a rather limited audience and is not accessible to people outside the community. The authors have replied to my comment and added a few remarks to the paper. While these comments have improved the manuscript, I am still not excited about it and still think the value of the work is not easily accessible for a general audience. But I can acknowledge that coherence at room temperature is a very crucial task for quantum technologies and that the research might spark experiments. Thus, it might fit into the scope of Nature Communications.

Reply

We would like to thank the reviewer for the thoughtful concerns and positive evaluation on our paper. We believe that our revised version is more accessible for the general audience and hope that our predictions would lead to further experimental work.

Response to Reviewer 3.

Comment

In the revised version of "Incoherent Nonadiabatic to Coherent Adiabatic Transition of Electron Transfer in Colloidal Quantum Dot Molecules", the authors provided further explanations to confirm the novelty. And I am satisfied with the author's response to my comments. And recommend the current version could be published in NATURE COMM without further revisions.

Reply

We would like to thank the reviewer for the thoughtful comments and suggestions on our original manuscript.